# Risk and Adversity Factors in Adult Patients with Comorbid Attention Deficit Hyperactivity Disorder (ADHD), Binge Eating Disorder (BED), and Borderline Personality Disorder (BPD): A Naturalistic Exploratory Study

**DOI:** 10.3390/brainsci13040669

**Published:** 2023-04-16

**Authors:** Derek Ryan, Joseph Sadek

**Affiliations:** 1Faculty of Medicine, Dalhousie University, Halifax, NS B3H 4R2, Canada; derek.ryan@dal.ca; 2Department of Psychiatry, Faculty of Medicine, Dalhousie University, Halifax, NS B3H 4R2, Canada

**Keywords:** attention deficit hyperactivity disorder, ADHD, borderline personality disorder, BPD, binge eating disorder, BED, impulsivity, psychosocial risk factors, comorbidities, clinical profile

## Abstract

This study was a retrospective pilot chart review of adult attention deficit hyperactivity disorder (ADHD) patients diagnosed with and without comorbid binge eating disorder (BED) and borderline personality disorder (BPD). ADHD research is critical because of its prevalence and persistence into adulthood. In the literature, ADHD, BED, and BPD are linked to an underlying impulsivity factor. This comparative study examined whether differences existed between patient groups concerning risk factors, comorbid disorders, and continuous performance test (CPT) cognitive scoring. The main goal was to find significant associations suggestive of correlations between specific factors and a principal diagnosis of ADHD with comorbid BED and BPD. Study participants were patients between 18 and 30 diagnosed by a psychiatrist in an outpatient clinic between June 2022 and December 2022. Both the control and comorbidity groups included 50 participants (N = 100). Patients were randomly chosen based on the chronological order of intake visit dates at the clinic and were selected as participants upon meeting the inclusion criteria. Data were collected through the Med Access EMR database, with quantitative data analyzed using SPSS and chi-squared *p*-value tests. The results showed significant associations between a principal diagnosis of ADHD with comorbid BPD and BED and (1) having four or more overall risk factors; (2) having five specific reported psychosocial risk factors: family issues, bullying, poverty, trouble with the law, and physical abuse; and (3) having on average more risk factors and comorbidities as compared to ADHD patients without comorbid BPD and BED. No association was found between low CPT scores and, thus, differential cognitive functionality between groups. This research will inform future study hypotheses to develop the clinical profile of ADHD patients with comorbid BED and BPD.

## 1. Introduction

### 1.1. Attention Deficit Hyperactivity Disorder (ADHD)

ADHD is a neurodevelopmental disorder characterized by impulsiveness, inattention, restlessness, and hyperactivity [1]. This classification is provided because onset occurs in early childhood before age 12 and is characterized by developmental deficits inconsistent with or excessive for developmental level or age. ADHD has a global prevalence rate of 5% in school children and 2.5% in adults, with the diagnosis being more prevalent in males [2]. Cultural and gender-related diagnostic issues have been identified as possible contributing factors to the heterogeneity in ADHD prevalence rates within populations and between regions [1]. For example, there are lower ADHD diagnostic rates within Latino and African American populations in the United States that may be related to cultural differences in informant symptom ratings [1]. Additionally, it has been found that inattentive ADHD presentation is more common in females than males, which may have contributed to historically lower rates of ADHD identification in females by clinicians [1]. As there are no biological tests to diagnose ADHD, varied clinical assessment methodologies may also give rise to these differences [3].

Three subtype specifications of ADHD have been previously identified: hyperactive-impulsive type, inattentive-distractible type, and combined type [4]. However, this terminology has been modified in the Diagnostic and Statistical Manual of Mental Disorders, Fifth Edition (DSM-5), which deemphasizes ADHD subtypes by classifying them as presentations [5]. This change agrees with current research demonstrating that ADHD symptoms are dynamic and susceptible to change across the lifespan as opposed to static qualities [5]. Thus, an individual’s ADHD presentation demonstrates current symptomology. A minimum of five symptoms in adults and six in children and adolescents of the DSM-5-specified symptoms for a given ADHD presentation must have been persistent over the past six months to meet the criteria [1]. Apart from these, diagnostic specifications for disease severity are included, ranging from mild to severe.

Inattention refers to lacking the ability to stay on task, listen when spoken to, organize tasks, or engage in mentally strenuous activities in addition to high distractibility and forgetfulness [1]. Hyperactivity pertains to excessive or inappropriate motor movement such as fidgeting, running, talking, interrupting, and waiting. In adulthood, this can manifest as inner restlessness [4]. Relevant to this study, impulsivity refers to behaving or acting in the moment without consideration, typically in an inappropriate or risky manner [5]. Although a level of impulsiveness is to be expected in normal individuals, when it begins to impact daily functioning (e.g., social or occupational), it crosses a threshold, becoming pathological. An example of this would be failing to assess long-term consequences while making decisions, demonstrating a desire for instant gratification [1]. All three ADHD presentations are common; however, hyperactive and impulsive symptoms typically diminish with age [5].

The interactions of many genetic and environmental risk factors are involved in the etiology of ADHD [3]. It is a complex disease that is multifactorial and strongly inherited within families. Exposure to non-genetic factors in the womb (prenatal), during birth (perinatal), and throughout childhood (psychosocial) have been associated with ADHD development. Moreover, the presence of comorbidities in clinical settings are numerous in individuals diagnosed with ADHD [1]. Notably, there is a regular intersection between childhood ADHD and “externalizing disorders” such as oppositional defiant disorder [3]. Additional neurocognitive, anxiety, personality, substance use, and eating disorders may exist.

### 1.2. Binge Eating Disorder (BED)

BED is an eating disorder characterized by recurrent food binges, with excessive caloric consumption and loss of control without subsequent compensatory behaviours [1]. Examples of these habits include self-induced vomiting, extreme exercise, and fasting [6,7]. It is the third main category of eating disorders listed in the DSM-5 and was previously classified as an eating disorder not otherwise specified (EDNOS) in the DSM-4. An episode of binge eating is defined as a larger-than-normal quantity of food being consumed in a discrete period during which one feels unable to keep from or stop eating [1]. Onset typically occurs later in adult life, and individuals with BED have comorbid psychological illness and obesity [8].

To be diagnosed, this behaviour must have occurred at least once per week for three months, with BED severity ranging from mild to severe, depending on the frequency of episodes per week. Additionally, individuals must have marked distress regarding the episodes, plus three of the following symptoms: eating when not physically hungry, eating more rapidly than usual, feeling guilty after eating, and preferring to eat alone or eat when not physically hungry. The global prevalence of BED is 0.9%, with the diagnosis being more prevalent in females. However, the gender ratio is more balanced in BED than in bulimia nervosa [1].

Similar to ADHD, BED’s etiology is thought to result from complex interactions between multiple genetic and non-genetic factors [7]. Emerging research has implicated neurobiological impairments in the development of the disease, specifically focusing on the emotional regulatory, inhibitory control, and reward processing domains. Hence, impulsivity has been proposed as one of the central risk factors for BED [6].

### 1.3. Borderline Personality Disorder (BPD)

BPD is a cluster B personality disorder characterized by an intense fear of abandonment, recurring suicidal thoughts or self-harm, paranoid ideation or dissociation, identity difficulties, chronic feelings of emptiness, impulsive behaviour, and unstable moods and relationships [4]. The point prevalence of BPD is estimated at 1% in community settings, increasing to 22% in outpatient clinical settings [9]. Around 75% of patients diagnosed with BPD are female [1]. As outlined by the DSM-5, for a diagnosis of BPD, patients must have a chronic pattern of functional impairment in addition to five of nine listed DSM-5 criteria, including risky behaviours, fear of abandonment, intense mood swings, and patterned unstable relationships [1,4]. Notably, binge eating is one of the impulsiveness criterion parts of the BPD symptom profile. If all criteria are met for both disorders, both diagnoses are given.

### 1.4. Rationale

To our knowledge, no published study compares ADHD patients with and without comorbid BED and BPD. ADHD research is critical because of its prevalence, persistence into adulthood, and adverse outcomes extending beyond the affected individual [3]. Prior studies have demonstrated associations between ADHD and BED [10,11,12,13,14], ADHD and BPD [15,16], as well as BED and BPD [17,18]. In the literature, impulsivity has been proposed as being associated with BPD and BED [18]. In ADHD diagnoses, there is also a significant impulsivity factor [19], thus suggesting a possible underlying link between the three psychiatric disorders. However, such connections have not yet been thoroughly tested or fully understood [6]. It has been suggested for future research to investigate in a clinical sample the relationship between ADHD and BED concerning impulsivity. Nazar et al. also proposed that future research should investigate the prognosis and course of eating disorders comorbid with ADHD compared to either diagnosis alone [20]. This study has the potential to provide valuable insight into and develop the clinical profile of ADHD patients with BED and BPD as a distinct subgroup. This is a critical area to explore because this information will help to inform on currently unclear areas of ADHD, BED, and BPD treatment as well as address and identify potential risk factors for the comorbid disorders.

## 2. Objectives

This pilot study aimed to describe the clinical profiles of adult ADHD patients diagnosed with comorbid BED and BPD and investigate existing differences in risk and adversity factors seen in these patients compared to adult ADHD patients without comorbid BED and BPD.

We hypothesized that ADHD patients diagnosed with comorbid BED and BPD would have an increased number of (1) overall risk factors, (2) specific perinatal and psychosocial factors, (3) psychiatric comorbidities, and (4) lower continuous performance test (CPT) scores as compared to ADHD patients without comorbid BED and BPD.

## 3. Materials and Methods

### 3.1. Study Design

This was a comparative retrospective pilot chart review of adult ADHD patients diagnosed with and without comorbid BPD and BED between June and December 2022. Patients of this clinic were provided informed consent upon intake regarding the use of their anonymized medical data and personal health information (PHI) in clinical research studies. Hence, no additional consent form was required. Patients were provided with the option to decline consent. A mechanism in the electronic medical record was used to flag those who had not agreed to the clinic’s informed consent form. The Nova Scotia Health Research Ethics Board provided ethics approval for the study (REB File #1028306). To minimize confounders in this study, participants were artificially matched between groups. This way, study participants with similar characteristics were classified within the same general age group. Sex comparisons were not made due to the limited sample size. The study’s methodology was used in previous studies [21].

### 3.2. Study Setting and Population

This study took place within a naturalistic outpatient psychiatric clinic setting. All study participants were patients at this clinic; the population comprised 100 patients, with 50 included in each study group. Thus, the study group had 50 patients diagnosed with ADHD comorbid with BED and BPD, whereas the control group consisted of 50 ADHD patients without comorbid BED and BPD. The sample was a convenience sample since this was a pilot study.

### 3.3. Inclusion and Exclusion Criteria

To be included in this study, participants were required to meet the inclusion criteria for either the comorbidity or control group. The inclusion criteria for the comorbidity group required patients to meet the DSM-5 criteria for BED and BPD as indicated by (1) having three or more risk factors for BED self-indicated on the Sadek Adult Assessment Questionnaire (SAAQ) or a confirmed diagnosis of ADHD by a psychiatrist or both; (2) having five or more risk factors for BPD self-indicated on the SAAQ or a confirmed diagnosis of ADHD by a psychiatrist or both; (3) as well as having a confirmed diagnosis of ADHD by a psychiatrist; and (4) being over the age of 18 at the time of initial intake. The inclusion criteria for the control group required patients not meet the DSM-5 criteria for BED and BPD as indicated by (1) having less than three risk factors for BED self-indicated on the SAAQ and no confirmed diagnosis of BED by a psychiatrist; (2) having less than five risk factors for BPD self-indicated on the SAAQ and no confirmed diagnosis of BPD by a psychiatrist; (3) as well as having a confirmed diagnosis of ADHD by a psychiatrist.

The exclusion criteria for this study were patients who (1) were under 18 at the time of initial intake; (2) had their initial intake after December 2022; or (3) did not provide consent to have their PHI used for clinical research purposes.

The first 50 patients meeting the specified inclusion criteria on initial assessment at the outpatient psychiatry clinic for either the comorbidity or control group were included in this study, working backwards chronologically from 28 December 2022. Patients were over 18 because, according to the literature, the onset of BED usually occurs later in life (Fairburn & Harrison, 2003). Thus, the presence of childhood trauma as a predictor of BED and ADHD together can be evaluated. The goal was to produce results that can lead to more robust clinical studies and the development of specific diagnostic criteria for a principal diagnosis of ADHD with comorbid BED and BPD; therefore, the sample size was sufficient for our purposes.

### 3.4. Data Collection and Analysis

Study participant PHI data were obtained from the clinic Med Access EMR database. Data collected included (1) patient demographic information (age and sex); (2) self-reported information from the Sadek Adult Assessment Questionnaire (SAAQ), including BED and BPD symptoms as well as perinatal, developmental, and psychosocial risk factors; (3) physician-reported information and patient psychiatric diagnoses (ADHD and relevant comorbidities); and (4) CPT scores. The SAAQ is a DSM-5-based questionnaire regarding risk and adversity factors as related to psychiatric comorbidities. All data collection was performed in a secure office on a password-protected computer. Furthermore, all data were de-identified when entering a password-encrypted Microsoft Excel spreadsheet.

The ten comorbid disorders investigated included BED, BPD, generalized anxiety disorder (GAD), major depressive disorder (MDD), obsessive-compulsive disorder (OCD), oppositional defiant disorder (ODD), specific learning disorder (SLD), substance use disorder (SUD), Tourette syndrome (TS), gambling addiction, anorexia nervosa (AN), and psychosis. There were nine reported risk factors investigated overall. The perinatal risk factors were complications during child delivery, including umbilical cord wrapping around the child’s neck, placement of the child in the neonatal intensive care unit (NICU), and low birth weight. The developmental risk factors included delayed achievement of walking and talking milestones. The psychosocial risk factors included having been bullied as a child, physically abused, sexually abused, having had family issues, growing up in poverty, having trouble with the law, and not graduating high school. Family issues were defined as the participant or physician reporting emotional abuse, the mother or father being unaffectionate or uncaring, and not getting along with siblings. Sexual abuse was specified as having occurred between the ages of 5 and 15. Trouble with the law was defined as the participant or physician reporting previous encounters with law enforcement, having stolen many times, or having broken into homes.

### 3.5. Statistical Analysis

Qualitative and quantitative data were electronically digitized in a Microsoft Excel version 16.2^®^ spreadsheet, tabulated, abstracted, and checked by two psychiatrists for processing and analysis. The IBM Statistical Package for the Social Sciences (SPSS), version 26.0, was used for all analyses. Descriptive statistics were completed where themes were identified in the study participants’ history for all relevant criteria, such as age, sex, and education. Mean and standard deviation (SD) analyses were completed for age, the number of risk factors, and comorbidities. Chi-squared association testing was used to assess categorical and non-parametric data analysis. Specifically, it was used to compare the reported specific and overall risk factors and CPT scores recorded in the PHI of study participants within and between the comorbidity and control groups. Using contingency tables, it was evaluated whether associations between the number of reported risk factors, specific reported risk factors, number of comorbidities, or CPT scores and a principal diagnosis of ADHD with comorbid BPD and BED were significant at the 0.05 level (*p* < 0.05).

## 4. Results

The age range of the study sample was 18 to 29, with a mean age of 23.95. Both the control and comorbidity groups had the same age range with a similar mean age. There was an even distribution of male compared to female participants in the study sample, with 25 female and 25 male participants in each group. Participants in the control group had a principal diagnosis of ADHD without comorbid BPD or BED. In contrast, participants in the comorbidity group all had a principal diagnosis of ADHD with comorbid BPD and BED. Most (92%) of the study participants had a high school education or higher, with 46% having completed undergraduate education. Between groups, participants had similar levels of education. However, only 2% of participants in the control group did not complete high school, as opposed to 8% in the comorbidity group. In both groups, an inattentive-type ADHD diagnosis was most common, with 62% of study participants diagnosed. There was a 22% rate of hyperactive-impulsive-type ADHD diagnosis in both groups. The study also recorded ten psychiatric comorbidities seen in study sample participants. GAD was the most common comorbidity of study participants (89%), and 100% of those in the comorbidity group were diagnosed with GAD as opposed to 78% in the control group. OCD, ODD, gambling addiction, and psychosis were comorbidities only seen in comorbidity group participants. AN, TS, and SLD were comorbidities only seen in control group participants. Comorbid MDD or SUD was observed in both study groups; however, 10% of participants in the comorbidity group had an SUD diagnosis in contrast with 2% of control participants. Table 1 describes the sociodemographic and diagnostic characteristics of the study sample.

The most reported risk factors by physicians and patients in the study sample were bullying (61%) and family issues (50%). Each risk factor was reported more commonly in comorbidity group participants, apart from having delayed milestones (walking or talking). However, participants in both groups reported similar rates of complicated births, including having an umbilical cord around the neck, low birth weight, and admission into the NICU (24% in the comorbidity group compared to 22% in the control group). In the comorbidity group, 78% of participants were reported as having been bullied, 72% had family issues, 36% had trouble with the law (theft, breaking and entering, police encounters), 36% lived in poverty, 30% were physically abused, 22% were sexually abused (between ages 5 and 15), and 8% did not graduate from high school. In the control group, 44% of participants were reported as having been bullied, 28% had family issues, 12% had trouble with the law, 8% lived in poverty, 8% were physically abused, 10% were sexually abused, and 2% did not graduate from high school. Table 2 describes the participant- and physician-reported risk factors in the sample population. Figure 1 displays the number of reported specific risk factors compared to the principal diagnosis.

The cognitive functional ability of study participants was assessed using continuous performance testing, which provides sustained auditory and visual attention quotient scoring. A visual or auditory score below 60 lies within the bottom 5th percentile and reflects extreme functional deficits. In the study sample, 65% of participants had a CPT score with an auditory or visual (or both) score below 60. In the control group, 68% of participants had such a score, compared to 62% of the comorbidity group participants. The rate of auditory and visual scores below 60 was similar between groups, with 46% of control group participants and 50% of comorbidity group participants having an auditory score below 60 and 52% of control group participants and 48% of comorbidity group participants having a visual score below 60. Table 3 describes the CPT scores with auditory and visual processing component scores lower than 60 in the study sample.

Participant CPT scores were dichotomized into two categories (scores below 60 and scores equal to or above 60) and grouped according to their control or comorbidity status. The study used the chi-squared *p*-value for statistical analysis. No significant association (*p*-value < 0.05) was found between participant CPT scores with an auditory or visual score below 60 and a principal diagnosis of ADHD with comorbid BPD and BED (*p* = 0.529368). Additionally, having an auditory score below 60 (*p* = 0.688921) or a visual score below 60 (*p* = 0.689157) was not significantly associated with a principal diagnosis of ADHD with comorbid BPD and BED. Table 4 records these non-significant associations.

Chi-squared testing was also used to assess the statistical association between specific risk factors and principal diagnosis. Contingency tables were made for each risk factor; the study categorized participants within the control and comorbidity groups as having or not having a specific risk factor. Five of the nine risk factors the study assessed showed statistical significance. Bullying (*p* = 0.000491), family issues (*p* = 0.000011), physical abuse (0.005048), poverty (*p* = 0.000726), and trouble with the law (*p* = 0.004958) were all risk factors significantly associated with a principal diagnosis of ADHD and comorbid BPD and BED. Exposure to the following risk factors, complicated birth (*p* = 0.812173), delayed milestones (*p* = 0.110041), sexual abuse (*p* = 0.101707), and not graduating high school (*p* = 0.307434), were not significantly associated with a principal diagnosis of ADHD and comorbid BPD and BED. Of note, combining physical and sexual abuse in childhood into a common ‘abuse’ category gave a significant association (*p* = 0.000365) to a principal diagnosis of ADHD and comorbid BPD and BED. Table 5 records these associations of specific risk factors compared to the principal diagnosis.

The study used chi-squared testing to analyze whether the total number of reported risk factors was associated with a principal diagnosis of ADHD with BED and BPD. To assess this, physician- and patient-reported risk factors were dichotomized into two categories: equal to or less than three and equal to or more than four. It was found that having equal to or greater than four risk factors was significantly associated with a principal diagnosis of ADHD with comorbid BPD and BED (*p* = 0.000051). In the comorbidity group, 68% of participants had equal to or greater than four risk factors compared to only 22% of participants in the control group. Table 6 demonstrates these findings.

The study calculated the mean number of risk factors for the control (2.46) and comorbidity (3.98) groups as well as the mean number of comorbidities (0.9 and 1.28, respectively). SD and standard error (SE) were also calculated, as reported in Table 7 and Figure 2, displaying error bars for each group that account for SE. It was found that, on average, participants in the comorbidity group with a principal diagnosis of ADHD and comorbid BPD and BED have significantly higher numbers of associated risk factors and other psychiatric comorbidities than the control group.

## 5. Discussion

The three main findings of this study were that a diagnosis of ADHD with comorbid BPD and BED in participants was (1) significantly associated with five reported specific risk factors: family issues, bullying, poverty, trouble with the law, and physical abuse; (2) significantly associated with having four or more risk factors; and (3) significantly associated with a higher mean number of risk factors and comorbid psychiatric disorders. A diagnosis of ADHD with comorbid BPD and BED was not significantly associated with lower CPT scores as compared to a diagnosis of ADHD in control group participants. Thus, the study suggests that patients who are exposed to more risk factors, or those who experience specific adversities in childhood (abuse, neglect, low socioeconomic status, trouble with the law), are more likely to develop ADHD with comorbid BPD and BED in adulthood. However, the CPT scores suggest similar cognitive functioning between comorbidity and control group participants.

To our knowledge, this is the first study to characterize specific features of a principal diagnosis of ADHD with comorbid BPD and BED. The study’s findings generally agreed with the literature surrounding these disorders. Rutter et al.’s model study comparing the prevalence of psychiatric disorders in two geographically separate child populations found that family-related environmental risk factors such as growing up socially disadvantaged, living in the foster system, and being exposed to parental criminality were all significantly associated with the presence of psychiatric disorders [22]. Although, the study findings suggested that it was the combination of and not the singular presence of specific risk factors that impaired mental development [23]. A follow-up study found a positive association between Rutter’s familial environmental risk factors and a risk for ADHD and its associated psychiatric, psychosocial, and cognitive impairments [24]. Notably, exposure to family issues was the risk factor most significantly associated with the diagnosis in the data set (*p* = 0.000011). These previous results align with the current study’s findings suggesting that family issues and poverty are risk factors significantly associated with a more severe diagnosis of ADHD with comorbid BPD and BED.

Additionally, participants with more risk factors were found more likely to have a principal diagnosis of ADHD with comorbid BPD and BED. This supports the finding that participants with a principal diagnosis of ADHD and comorbid BPD and BED had, on average, more risk factors than control group participants. Interestingly, another study investigating impulsivity, a factor related to ADHD, BPD, and BED, found that an increased number of impulsivity symptoms was correlated with more comorbidities [19]. This may help explain why, on average, comorbidity group participants had more psychiatric comorbidities than the control group participants.

Another preliminary study observed significantly higher incidences of ADHD in children who had suffered maltreatment and emotional trauma [25]. This was somewhat replicated in a 2013 study that found hostile parenting styles being associated with child ADHD symptoms [26]. Of interest, increased rates of post-traumatic stress disorder (PTSD) were also seen in these children with adverse experiences [25]. The authors suggest this may present clinically as BPD when in its chronic state. These study results are supported by recent meta-analyses findings, demonstrating that bullying-related trust issues and maladaptive parenting are risk factors for developing BPD [27]. These findings are supported by this study’s results that demonstrate significant associations between a principal diagnosis of ADHD with comorbid BPD and BED and exposure to bullying, family issues, and physical abuse in childhood as psychosocial risk factors. However, this study did not find sexual abuse as significantly associated with the comorbidity group diagnosis. This may be attributed to the limited sample size of this study or that sexual abuse is an underreported risk factor by participants.

Furthermore, symptoms of children with traumatic experiences sometimes match the clinical presentations of ADHD including inattention, hyperactivity, and impulsiveness [23]. This may mask the presence of an existing personality or anxiety disorder. When physical and sexual abuse were combined into a single category, childhood abuse was found to be significantly associated with ADHD and comorbid BPD and BED. However, this aggregate finding in a limited sample size is more susceptible to type 1 error or random chance.

The study findings also support the results of studies investigating environmental risk factors for eating disorders, specifically BED. Hilbert et al. found that childhood bullying and teasing were associated with an increased risk of BED, supporting the significant association between bullying in childhood and an ADHD diagnosis with comorbid BPD and BED [28,29]. Conduct issues were also found to be associated with BED. Externalizing disorders have been linked to ADHD diagnoses [1,23], therefore supporting a possible connection between the two conditions. Moreover, temperamental impulsivity was associated with an increased risk of BED, a known factor in ADHD hyperactive-impulsive presentation and BPD diagnoses [1]. This further suggests that specific psychosocial risk factors may predispose individuals to ADHD with BPD and BED comorbidities. Another study with a case-control design exploring the relationship between prior life events and BED found that the odds of developing BED were positively associated with the frequency of adverse events experienced [29]. Risk factors, including physical abuse, perceived risk of physical abuse, safety concerns, stress, and criticisms were all more common in the BED patient group [29], thus further supporting this study’s associations of such specific risk factors to ADHD with comorbid BPD and BED.

Uniquely, this study demonstrates a possible significant association between being in trouble with the law and a principal diagnosis of ADHD with comorbid BPD and BED. Prior studies have found ADHD-related hyperactive-impulsivity and externalizing factors in childhood to be predictors of future adult criminality [30]. However, in a recent study, it was found that impulsivity, and not hyperactivity, was the specific criminogenic factor [31]. This is interesting, as there has not been much research surrounding criminality and the underlying impulsivity factor related to ADHD, BED, and BPD. Few studies have investigated the association between BPD symptoms and criminality, even though it has been linked to violence and aggressiveness [32]. Therefore, this study finding suggests that investigating impulsivity and its interactions in ADHD with comorbid BPD and BED may help elucidate predictable patient behaviours.

Interestingly, most study participants, including those within the comorbidity group, were diagnosed with ADHD, predominantly inattentive type, as seen in Table 1. Based on impulsivity being the common factor seen in ADHD, BPD, and BED diagnoses [18,19], we had expected ADHD, predominantly hyperactive-impulsive type, to be the most common diagnosis amongst patients with BPD and BED comorbidities. Possible explanations for this discrepancy include the fact that study participants were adults at the initial clinic intake date; studies have shown that hyperactive and impulsive symptoms are more prevalent in childhood and show an age-dependent decline at a faster rate than inattentive symptoms [33]. This supports the semantic change from ADHD ‘subtype’ to ‘presentation’ made in the diagnostic criteria for the disorder in the DSM-5, reflecting its dynamic nature [5]. As this study did not collect participants’ past medical history, it is plausible that some may have presented with hyperactive and impulsive symptoms in childhood, relevant to their BPD and BED comorbidities, that were ignored and have since diminished.

CPT scores, and thus cognitive functionality in terms of auditory and visual processing, were not significantly different between control and comorbidity group participants. Furthermore, perinatal and developmental risk factors were seen at similar rates between groups, as was the psychosocial factor of not graduating from high school. The lack of significant associations between ADHD with comorbid BPD and BED and these variables suggests that ADHD may be the only diagnostic factor contributing to them.

### Limitations and Future Directions

This study’s small sample size (N = 100) limits the validity of the main findings. Thus, the significant associations found between a diagnosis of ADHD with comorbid BPD and BED and (1) overall risk factors, (2) specific risk factors, and (3) overall comorbidities cannot be interpreted as definitive correlations. Regardless, the results provide interesting signals that certain psychosocial factors, namely bullying, family issues, physical abuse, poverty, and trouble with the law, are associated with this principal diagnosis. Additionally, the results suggest that the higher number of risk and adversity factors and comorbidities one has, the more likely one will be diagnosed with ADHD and comorbid BED and BPD. Future studies may take these findings as a priori hypotheses, which, by using power calculations to determine an appropriate participant sample size, may validate the findings of this study. Future studies may also investigate the gender and age differences seen in ADHD, BED, and BPD diagnoses with respect to risk and adversity factors and comorbidities.

Furthermore, this study could not collect PHI related to family medical history, prior management plans, or medication regimens. There is evidence for familial and genetic factors playing roles in the susceptibility to and development of ADHD, BPD, and BED [23,27,34]. Not discussed in this paper, gene-environment correlations have been made, for example, in animal studies, where it has been demonstrated that specific gene variants in certain environmental conditions can elicit varying behaviours [3]. This suggests that genetic and environmental factors play dual roles in developing psychiatric disorders. It is not as simple as one psychosocial risk factor aggravating the development of ADHD with or without BPD and BED. Individuals may have varying genetic susceptibilities that could be activated in certain environmental conditions [3].

Recall bias could contribute to some inaccuracy in the reports, but patients affirmed that they provided the information to the best of their abilities.

## 6. Conclusions

The effects of risk factors on ADHD, BPD, and BED development are essential to help understand the clinical profiles of patients within this specific ADHD population. With more robust clinical understanding of this diagnosis, clinicians may be able to diagnose and treat ADHD patients with comorbid BPD and BED more effectively. More research is needed in this area. Patients with the diagnosis of ADHD with BPD and BED struggle with several symptoms that may significantly limit their abilities to have a full range of social, occupational, or academic level of functioning. Research that facilitates the diagnosis and management of these disorders may improve the quality of life of patients who struggle with these disorders. Further specific research in females with these disorders is needed since the diagnosis of ADHD in middle-aged females has been overlooked for years. The next steps in research should also include gender and age differences in ADHD/BED/BPD prevalence rates/comorbidities.

## Figures and Tables

**Figure 1 brainsci-13-00669-f001:**
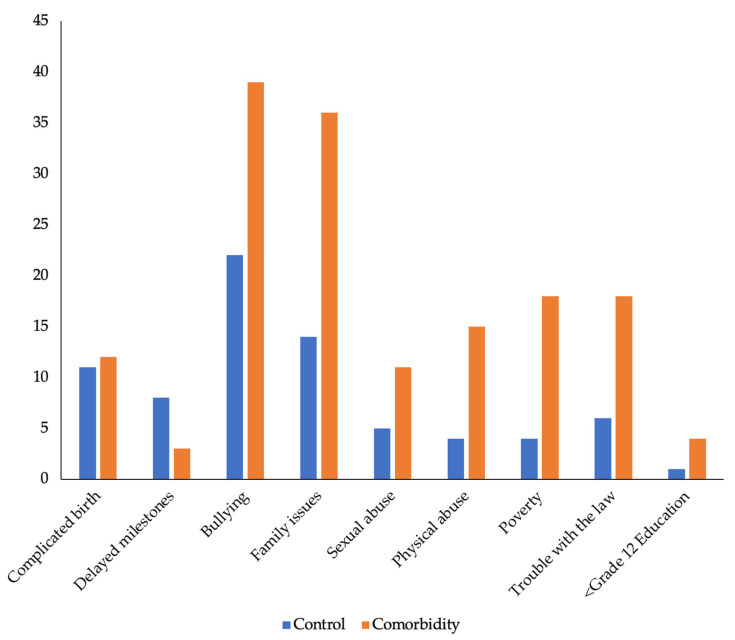
Number of reported specific risk factors as compared to principal diagnosis.

**Figure 2 brainsci-13-00669-f002:**
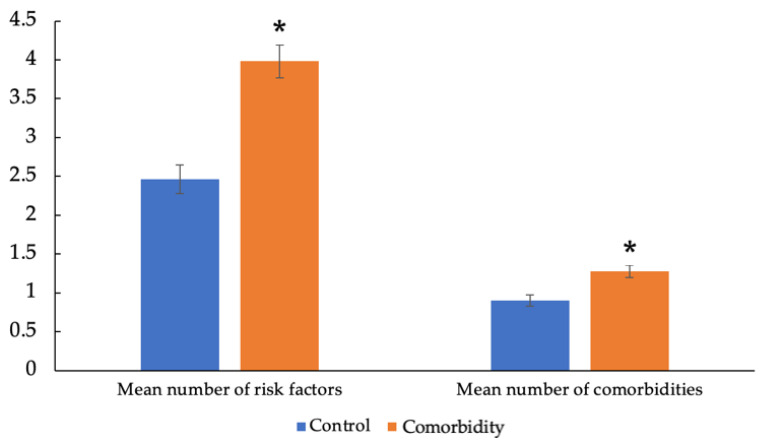
Mean number of risk factors and comorbidities compared to principal diagnosis. *: Indicates significant difference between the standard error of the means.

**Table 1 brainsci-13-00669-t001:** Sociodemographic and diagnostic characteristics of the study sample (N = 100).

Characteristic	Study (N = 100)	Control (N = 50)	Comorbidity (N = 50)
**Age**			
Range	18–29	18–29	18–29
Mean	23.95	23.56	24.34
SD	3.36	3.30	3.42
**Education**			
<Grade 12	5	1	4
Grade 12	49	24	25
Undergraduate	43	23	20
Postgraduate	3	2	1
**Sex**			
Male	50	25	25
Female	50	25	25
**Major Diagnosis**			
ADHD	50	50	0
ADHD, BPD, BED	50	0	50
**ADHD Presentation**			
Inattentive	62	30	32
Hyperactive-impulsive	22	11	11
Combined	7	5	2
Unspecified	9	4	5
**Comorbid Disorders**			
Generalized anxiety disorder (GAD)	89	39	50
Substance use disorder (SUD)	6	1	5
Obsessive compulsive disorder (OCD)	4	0	4
Major depressive disorder (MDD)	4	2	2
Specific learning disorder (SLD)	1	1	0
Oppositional defiant disorder (ODD)	1	0	1
Tourette syndrome (TS)	1	1	0
Gambling addiction	1	0	1
Anorexia nervosa (AN)	1	1	0
Psychosis	1	0	1

**Table 2 brainsci-13-00669-t002:** Participant- and physician-reported risk factors of the study sample (N = 100).

Risk Factor	Study (N = 100)	Control (N = 50)	Comorbidity (N = 50)
Bullying	61	22	39
Family issues	50	14	36
Trouble with the law	24	6	18
Complicated birth	23	11	12
Poverty	22	4	18
Physical abuse	19	4	15
Sexual abuse	16	5	11
Delayed milestones	11	8	3
<Grade 12 education	5	1	4

**Table 3 brainsci-13-00669-t003:** CPT scores with auditory and visual component scores less than 60 in the study sample (N = 100).

CPT Scoring	Study (N = 100)	Control (N = 100)	Comorbidity (N = 50)
CPT score < 60	65	34	31
Auditory score < 60	48	23	25
Visual score < 60	50	26	24

**Table 4 brainsci-13-00669-t004:** Association of overall CPT, auditory, or visual component scores below 60 as compared to principal diagnosis.

CPT Scoring	Control (N = 100)	Comorbidity (N = 50)	*p*-Value
CPT score < 60	34	31	0.529368
Auditory score < 60	23	25	0.688921
Visual score < 60	26	24	0.689157

**Table 5 brainsci-13-00669-t005:** Association of specific risk factors as compared to principal diagnosis.

Risk Factor	Control (N = 50)	Comorbidity (N = 50)	*p*-Value
Bullying	22	39	**0.000491**
Family issues	14	36	**0.000011**
Trouble with the law	6	18	**0.004958**
Complicated birth	11	12	0.812173
Poverty	4	18	**0.000726**
Physical abuse	4	15	**0.005048**
Sexual abuse	5	11	0.101707
Delayed milestones	8	3	0.110041
<Grade 12 education	1	4	0.307434

**Table 6 brainsci-13-00669-t006:** Association of number of risk factors as compared to principal diagnosis.

Overall Risk Factors	Control (N = 50)	Comorbidity (N = 50)	*p*-Value
</=3 risk factors	39	19	
>/=4 risk factors	11	31	**0.000051**

**Table 7 brainsci-13-00669-t007:** Mean number of risk factors and comorbidities of participants in the control and comorbidity groups with standard deviation and standard error calculations.

Statistical Measure	Control (N = 50) Risk Factors	Comorbidities	Comorbidity (N = 50) Risk Factors	Comorbidities
Mean	2.46	0.9	3.98	1.28
SD	1.29693516	0.50507627	1.47758766	0.57285536
SE	0.18341433	0.20896245	0.07142857	0.08101398

## Data Availability

The data presented in this study are available on request from the corresponding authors.

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
