# Peer review of "Risk and Adversity Factors in Adult Patients with Comorbid Attention Deficit Hyperactivity Disorder (ADHD), Binge Eating Disorder (BED), and Borderline Personality Disorder (BPD): A Naturalistic Exploratory Study"

_brainsci, 2023, doi:10.3390/brainsci13040669_

Round 1

Reviewer 1 Report

I congratulate the authors for the presentation of the work.  I consider it to be relevant, interesting and timely.  I think it is well developed and argued, making clear the central elements of the research.  Perhaps it would be appropriate to further develop the conclusion as well as to differentiate a separate section dedicated to the limitations of the research.

Best regards.

Author Response

Response to Reviewer 1

Dear Reviewer,  

Thanks so much for your time, effort, and expertise in reviewing our manuscript. We are extremely grateful for your work. Your suggestions are helpful. We have added a separate section on limitations as instructed and developed the conclusion further. 

Thanks again.

Truly.,

Joseph Sadek and Derek Ryan

Reviewer 2 Report

The comparative study reviewed charts of 100 psychiatric patients diagnosed with adult attention deficit-hyperactivity disorder (ADHD) with and without comorbid binge eating disorder (BED) and borderline personality disorder (BPD). All three conditions are linked to an underlying impulsivity factor. The study investigated whether there were differences in comorbid disorders, risk factors, and continuous performance test (CPT) cognitive scoring comorbid disorders. Significant relationships were found for ADHD with BED and BPD comorbidities in terms of 1) psychosocial risk factors: family issues, bullying, poverty, trouble with the law, and physical abuse; 2) having four or more overall risk factors; 3) having on average more risk factors and comorbidities as compared to ADHD patients without comorbid BPD and BED.

Comment 1: “Cultural and gender-related diagnostic issues have been identified as possible contributing factors to the heterogeneity in ADHD prevalence rates within populations and between regions [1]”. Is this pointing to recent reports of more adult women being diagnosed with ADHD than previously? Perhaps it may help to provide an example, that females were being overlooked for the ADHD in their formative years several decades ago.

Comment 2: With regards to this last comment and the higher prevalence rates noted in Section 1.2/1.3 for BPD and BED, it appears that a window of opportunity for early intervention has been possibly missed because of ADHD was thought to be more of male concern. These comorbidities may be exacerbated by the absence of ADHD intervention. “However, there is currently no published study comparing ADHD patients with and without comorbid BED and BPD” … “This study has the potential to provide valuable insight into and develop the clinical profiles of ADHD patients with BED and BPD as a distinct sub-group”. Is it too much of a risk at this point to mention how females appear to have been overlooked and the higher rates of diagnosis in middle-age for ADHD should also be investigated for subgroup (BED and BPD). Perhaps it is not possible for the current study to extract this information, but this seems to be subtly referred to and not clearly outlined what the intent is.

Comment 3: Table 1 presents the 50/50 representation of male to females in the study. However, the results are not delineated for each sex. Rather, they are grouped together. The age range being 18-29 means my previous comment about middle-aged women (being diagnosed with ADHD a higher rates than before) is not relevant to the current study’s data.  However, I think there could be more made of interpreting data in the future, noting the gender and age differences in ADHD/BED/BPD prevalence rates/comorbidities.

Only minor spelling/grammar check required

Author Response

Reviewer #2

Response to Reviewer #2

Dear Reviewer,  

Thanks so much for your time, effort, and expertise in reviewing our manuscript. We are extremely grateful for your work. Your suggestions are helpful. Please see our response to your helpful comments below.

Thanks again

Truly.,

Joseph sadek and Derek Ryan

Comment 1: “Cultural and gender-related diagnostic issues have been identified as possible contributing factors to the heterogeneity in ADHD prevalence rates within populations and between regions [1]”. Is this pointing to recent reports of more adult women being diagnosed with ADHD than previously? Perhaps it may help to provide an example, that females were being overlooked for the ADHD in their formative years several decades ago.

1. Response: We completely agree. We have added a reflection in the background to highlight that females were being overlooked for the ADHD in their formative years several decades ago. Now the diagnosis of ADHD in females is recognized and supported.

Comment 2: With regards to this last comment and the higher prevalence rates noted in Section 1.2/1.3 for BPD and BED, it appears that a window of opportunity for early intervention has been possibly missed because of ADHD was thought to be more of male concern. These comorbidities may be exacerbated by the absence of ADHD intervention. “However, there is currently no published study comparing ADHD patients with and without comorbid BED and BPD” … “This study has the potential to provide valuable insight into and develop the clinical profiles of ADHD patients with BED and BPD as a distinct sub-group”.

Is it too much of a risk at this point to mention how females appear to have been overlooked and the higher rates of diagnosis in middle-age for ADHD should also be investigated for subgroup (BED and BPD). Perhaps it is not possible for the current study to extract this information, but this seems to be subtly referred to and not clearly outlined what the intent is.

2. Response: We agree with the reviewer’s comment about females appear to have been overlooked and the higher rates of diagnosis in middle-age for ADHD should also be investigated for subgroup (BED and BPD).

This was added in the conclusion section to suggest future larger research in this area.

Comment 3: Table 1 presents the 50/50 representation of male to females in the study. However, the results are not delineated for each sex. Rather, they are grouped together. The age range being 18-29 means my previous comment about middle-aged women (being diagnosed with ADHD a higher rates than before) is not relevant to the current study’s data.  However, I think there could be more made of interpreting data in the future, noting the gender and age differences in ADHD/BED/BPD prevalence rates/comorbidities.

3. Response: We completely agree there could be more made of interpreting larger data in the future to assess the gender and age differences in ADHD/BED/BPD prevalence rates/comorbidities.

We updated the conclusion to reflect that as well.

Thanks again for excellent suggestions.

Joseph Sadek and Derek Ryan